# Identification of *miR397a* and Its Functional Characterization in Callus Growth and Development by Regulating Its Target in *Liriodendron*

**Dan Wang** [1], **Fengjuan Lu** [1], **Ye Lu** [1], **Tielong Cheng** [2], **Jisen Shi** [1], **Jinhui Chen** [1] and **Zhaodong Hao** [1,*]

[1]  Co-Innovation Center for Sustainable Forestry in Southern China, Key Laboratory of Forest Genetics & Biotechnology of Ministry of Education of China, Nanjing Forestry University, Nanjing 210037, China; wangdnjfu@hotmail.com (D.W.); 13675160925@163.com (F.L.); luye@njfu.edu.cn (Y.L.); jshi@njfu.edu.cn (J.S.); chenjh@njfu.edu.cn (J.C.)
[2]  College of Biology and the Environment, Nanjing Forestry University, Nanjing 210037, China; chengtl@njfu.edu.cn
*  Correspondence: haozd@njfu.edu.cn

**Abstract:** Callus growth and development, a crucial process in plant propagation, is involved in hormonal balance and abundant gene regulation. MiRNAs are key regulators in the process of cell differentiation and development. MiR397 was identified as participating in plant growth, development, and response to stress, and it was regulated by targeting the *LAC* gene. The regulatory function of miR397 during callus growth and development was not clear in *Liriodendron*. In this study, LhmiR397a and its targets were identified, and its regulatory function between LhmiR397a and *LhLAC11* was shown using qRT-PCR and transient expression in protoplasts. Furthermore, to clarify the regulatory function of LhmiR397a-*LhLAC11*, transgenic calli overexpressing *LhMIR397a*, *LhLAC11*, and *mLhLAC11* were separately obtained by *Agrobacterium*-mediated transfer. The results showed that overexpressing *LhMIR397a* might retard callus proliferation, while overexpressing LhLAC11 or mLhLAC11 could promote callus proliferation. Genes associated with the cell cycle had decreased expression when LhMIR397a was overexpressed, while increased expression was observed when LhLAC11 or mLhLAC11 was overexpressed. Additionally, the calli overexpressed with *LhMIR397a* could generate early cotyledons 21 days after induction, and the somatic embryo induction time was short compared with other genotypes. This study identified LhmiR397a and its targets and provided a functional characterization of LhmiR397a in callus growth and development by regulating its target in *Liriodendron*.

**Keywords:** miR397a; laccase gene; callus; proliferation; cell cycle; development

## 1. Introduction

*Liriodendron sino-americanum* is an excellent woody tree species and garden afforestation tree species. It is crucial to improve the quality and yield of *Liriodendron sino-americanum*. According to previous studies, the somatic embryogenesis system is a powerful tool for massive propagation and character improvement of higher plants. As a result, the determination of the mechanism of somatic embryogenesis in *Liriodendron sino-americanum* is urgently needed.

Having pluripotency for regenerating new organs or whole plants under certain conditions is one central characteristic of plant cells [1]. The in vitro plant regeneration process often begins with callus induction. Callus refers to a disorganized cell mass and can be induced by wounding stress, which activates a set of reprogramming-related genes, including *ERF115*, *WIND1–WIND4*, *PLT3*, *PLT5*, and *PLT7* [2–4]. In addition, calli can be formed in vitro when cultured with an optimal ratio of auxin/cytokinin [5]. In *Liriodendron*, a balance between auxin and cytokinin successfully induced callus formation [6].

Somatic embryogenesis (SE) is the process of embryogenesis induced by somatic cells. Under appropriate conditions, calli can be induced to undergo the SE program [7]. The in vitro regeneration process was mainly regulated by auxin associated with cytokinin; a low auxin–cytokinin ratio could induce shoot regeneration, and a high auxin–cytokinin ratio could promote root formation [8,9]. In addition to phytohormones as regulatory factors, multiple genes were employed during the SE process. MicroRNAs (miRNAs) are a class of endogenous, noncoding, small RNAs of only 17–25 nucleotides (nt) in length [10,11]. Previous studies have demonstrated that miRNAs are essential for cell differentiation and the development process during early embryogenesis [12–14]. The role of miRNAs during somatic embryogenesis has been verified in multiple plants, including loblolly pine [15], *Arabidopsis* [16], sweet orange [17], larch [18], longan [19], cotton [20], radish [21], and *Lilium* [22]. Several miRNAs, such as miR156, miR157, miR159, miR160, miR165, miR166, miR167, miR390, miR393, and miR396, have emerged as key regulators in embryogenesis.

MiR397 was first identified as a novel member of the abiotic-stress-regulated miRNAs in *Arabidopsis* [23]. Later, several studies confirmed the stress responsive regulator of miR397 in other plants. In *Arabidopsis* [24] and *Populus trichocarpa* [25], miR397 was reported to be a Cu-responsive miRNA. In *Arabidopsis* [26] and maize [27], miR397 was identified as a regulator of nitrogen starvation. Dong [28] reported that miR397 overexpression improved plant tolerance to cold stress in *Arabidopsis*. The regulation of drought response in *Oryza sativa* was also confirmed [29]. In addition, in rice, Luo [30] identified that miR397 carries out a vital regulated role of maintaining embryonic cells in a thin wall and meristematic state. It also has a regulatory role in flowering and flower, seed, and fruit development in *Arabidopsis* [31,32], rice [33], *Nicotiana tabacum* [34], citrus [35], and pear [36]. Recently, the biological functions of miR397 were broadly involved in plant growth, development, and stress response [37].

In our previous study, we found that miR397 can be tested by chip hybridization, and the targeted transcripts of miR397 were identified by the degradome sequence. However, the regulatory effect of miR397 during callus development and differentiation in *Liriodendron sino-americanum* remains unknown. In this study, LhmiR397a was identified, and the sequences of LhmiR397a and its targets were obtained. Additionally, the regulatory role of LhmiR397a and its targets were further analyzed.

## 2. Materials and Methods

### 2.1. Plant Materials

Synchronized cultures of somatic embryos at different developmental stages were obtained from an improved *L. sino-americanum* system (*L. tulipifera* × *L. chinense*) established in our lab. They were immediately frozen in liquid nitrogen and stored at −80 °C for further use according to previous methods [6].

### 2.2. Bioinformatics Analysis

The miR397 sequence was analyzed according to its miRNA sequence and microarray as previously reported. The miR397 sequence in other plants was downloaded from miRbase22.1 (http://www.mirbase.org/ftp.shtml, accessed on 30 June 2021). WebLogo 2.8.2 (http://weblogo.berkeley.edu/logo.cgi, accessed on 30 June 2021) [38] was used to analyze sequence conservation. The secondary structure was analyzed with an RNAfold 2.4.18 (http://rna.tbi.univie.ac.at/cgi-bin/RNAWebSuite/RNAfold.cgi, accessed on 30 June 2021) [39]. The targets of miR397a were predicted with psRNATarget 2017 (http://plantgrn.noble.org/psRNATarget/, accessed on 30 June 2021) [40] and analyzed using a t-plots assay with the data reported previously.

### 2.3. Cloning and Transformation

The pre-miR397a cDNA was amplified from *L. chinense* (Lushan) by PCR with the forward primer 5′ GTG ATG AGG ACG GTG TCG GTC TTG 3′ and reverse primer 5′ ATT TGG CGC GCT ATC TAT CAT GC 3′. *LhLAC11* cDNA was amplified with the forward

primer 5′ AGA CAT GGA GTC ATG GGT TCG AG 3′ and reverse primer 5′ TCG GTC CAT CCT CAT CAT CTA TTG 3′. The mutagenic primer for mutant *LhLAC11* was 5′ AAG AAG AGC TCT TCG TTT AGG GCG GCG TTA ATG AGC CGG AGC ATG 3′. The transient transfection vectors were constructed with the pJIT166 plasmid and transformed into *Liriodendron* protoplasts mediated by PEG. The protoplast was prepared based on a previous report [41]. The plant expression vectors were constructed with the pBI121 plasmid and transformed into *Liriodendron* calli by *Agrobacterium*-mediated transfer.

### 2.4. Quantitative Real-Time PCR

The total RNA of each sample was extracted using the Total RNA Purification kit (Norgen Biotek Corporation, Thorold, ON, Canada). The reverse transcription reactions for miRNAs and cDNA were carried out using the miRNA 1st strand cDNA Synthesis kit (by stem-loop) (Vazyme, Nanjing, Jiangsu, China) and the HiScript II Q RT SuperMix for qPCR (Vazyme, Nanjing, Jiangsu, China), respectively. Quantitative real-time PCR (qRT-PCR) was performed on the Roche Applied Science LightCycler 480. qRT-PCR was performed as described by Li et al. [6]. The expression level of *18S* and *Actin* were used as the internal control to standardize the RNA samples for each reaction. The relative gene expression levels were calculated using the $2^{-\Delta\Delta}$ method, and error bars indicate the standard deviation of three sample replicates. The stem-loop primer for miR397 was 5′ CTC AAC TGG TGT CGT GGA GTC CGG CAA TTC AGT TGA ATC AAC GC 3′. The qRT-PCR primers for miR397 were a forward primer, 5′ ACA CTC CAG CTG GGT CAT TGA GTG 3′, and a reverse primer, 5′ AAC TGG TGT CGT GGA G 3′. The qRT-PCR primers for *LhLAC11* were a forward primer, 5′ TGG GCC ACT CTC AAA TTG CT 3′, and a reverse primer, 5′ TAG GAG GAC GTT GGT GGT CT 3′. The qRT-PCR primers for *CYCA* were a forward primer, 5′ ATG CAA TGA ACA GGC AAC GG 3′, and a reverse primer, 5′ TTG GTG GTT GGT GCT GTC AT 3′. The qRT-PCR primers for *CYCB* were a forward primer, 5′ AGA GCT TGT GGA CTG TGC AA 3′, and a reverse primer, 5′ CAG CTC CGC GTA GAG GAT TC 3′. The qRT-PCR primers for *CYCD* were a forward primer, 5′ GTC GGA GCT CTT TCA GAC CC 3′, and a reverse primer. 5′ TGC AGT CGC GCT CTT CCA TTG TCT TG 3′. The qRT-PCR primers for *18S* were a forward primer, 5′ ATT TCT GCC CTA TCA ACT TTC G 3′, and a reverse primer, 5′ TTG TTA TTT ATT GTC ACT ACC TCC C 3′. The qRT-PCR primers for *Actin* were a forward primer, 5′ TGG ACT CTG GGG ATG GTG TTA′, and a reverse primer, 5′ CTC GGC TGT GGT TGT GAA AG 3′.

### 2.5. Callus Proliferation and MTS Assays

Calli from the wild-type and those overexpressing *LhMIR397a*, *LhLAC11*, or *mLhLAC11* were cultured on callus-inducing medium. After culturing for 25 days, the fresh calli were weighed. The cell proliferation rate was tested with an MTS assay using the Cell-Titer 96 Aqueous One Solution Cell Proliferation Assay Kit (Promega) as described previously [42].

### 2.6. Physiological Analyses

The soluble sugar, starch, and soluble protein content in different tissues were determined according to the method of Peng [43]. Fresh tissues (0.2–0.5 g) were obtained to detect and analyze biochemical events, with three replications performed per treatment.

## 3. Results

### 3.1. Identification of miR397 and LhmiR397a Differentially Expressed during Early SE

According to the miRNA sequence and microarray, we obtained four miR397 in *L. sino-americanum*. The miR397 in *Liriodendron* was 21 nt in length, which was consistent with that of most plants. The mature sequences of all four miR397s were compared with 84 mature miR397 sequences found in miRbase 22, and the results showed high conservatism among *Picea abies*, *Amborella trichopoda*, *Camelina sativa*, seven monocotyledons, and 23 dicotyledons. The 1st, 2nd, 14th, and 21st bases were completely conserved, while the bases at positions 5, 7, 9, 10, 11, and 15 were different (Figure 1A). To further iden-

tify LhmiR397a, the sequence of *pre-miR397a* was obtained with BLAST; the secondary structure was predicted with RNAfold. The results showed that the *pre-miR397a* might fold into a stem-loop structure (Figure 1B), which was considered a classical structure of pre-miRNA. The mature LhmiR397a was generated from the 5p arm of the precursor. In addition, the minimum folding free energy ($\Delta G$), which is considered a threshold for assessing the identity of pre-miRNA, was also evaluated. In this study, $\Delta G$ for *LhMIR397a* was $-49.41$ kcal mol$^{-1}$, which is within the standard range.

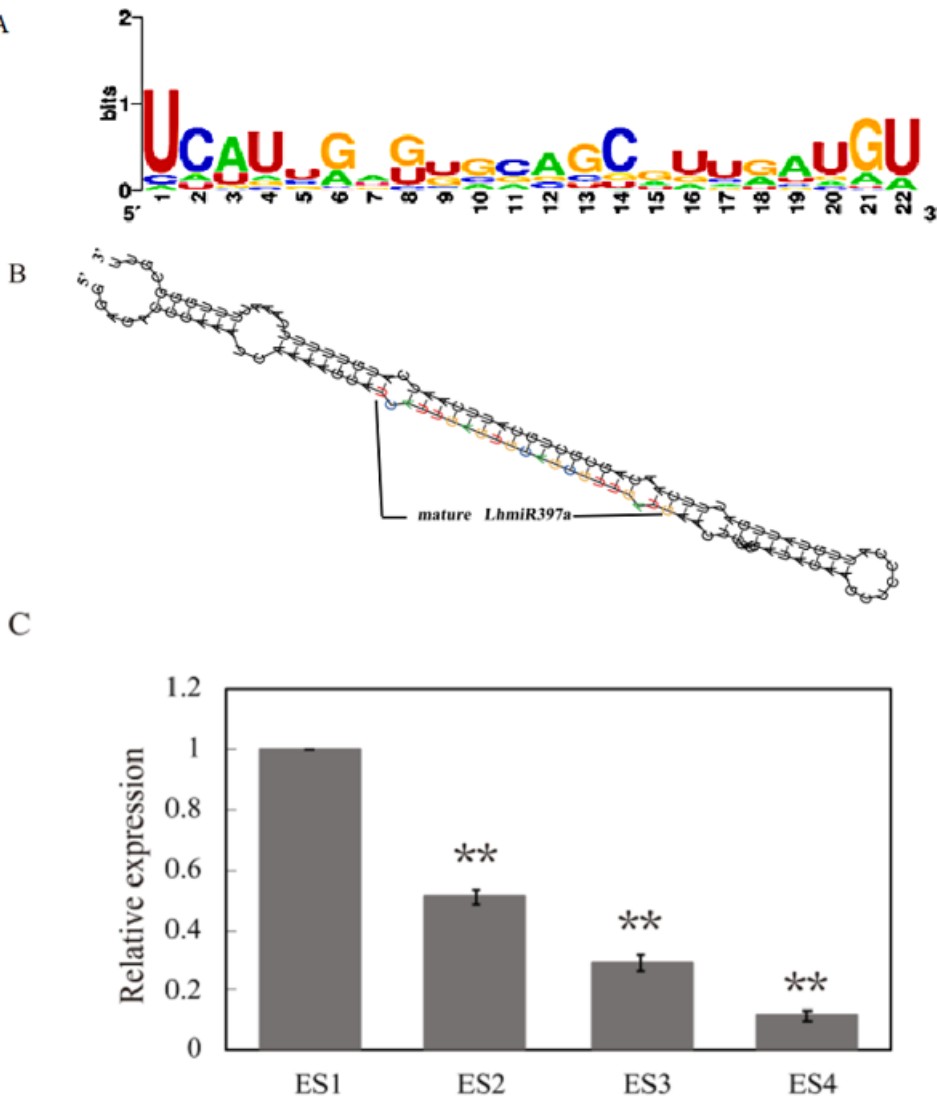

**Figure 1.** Analysis of the LhmiR397a sequence and its expression. (**A**) Analysis of sequence conservation in the miR397 family (weblogo.berkeley.edu, accessed on 15 June). (**B**) Hairpin structure of LhmiR397a precursors. (**C**) LhmiR397a expression level analyzed by qRT-PCR. Asterisks indicate statistically significant differences compared with ES1 by Student's *t* test (** $p \leq 0.01$).

The microarray assay indicated that LhmiR397a was differentially expressed during somatic embryogenesis progress. To confirm the effect of LhmiR397a on early SE, qRT-PCR was used to test the expression level at different induction times. The results demonstrated that the expression of LhmiR397a decreased during early SE (Figure 1C), and the trend was consistent with the microarray. The expression level of LhmiR397a in the embryonic suspension cells 7 days after sub-culture (ES1) was the highest. It was then reduced in embryonic single cells 2 days after sub-culture (ES2) and further reduced in embryonic tissues 2 days after differentiation culture (ES3). It reached the lowest expression in globular

embryos 7 days after differentiation culture (ES4). The results suggested that LhmiR397a may play an essential role in the early regulation of SE.

### 3.2. Three LAC Genes Are Targets of miR397a

Previously, studies showed that the laccase (LAC) gene is the target transcript of miR397a in other plants, such as *Arabidopsis*, rice, *Poplar*, pears, tobacco, and *Eucalyptus grandis*. In *L. sino-americanum*, a strongly matched position in *LhLAC* was identified for LhmiR397a by PsRNATarget prediction. Based on the degradome sequence, cleavage on *LhLAC* was confirmed at the 10th and 11th bases (Figure 2).

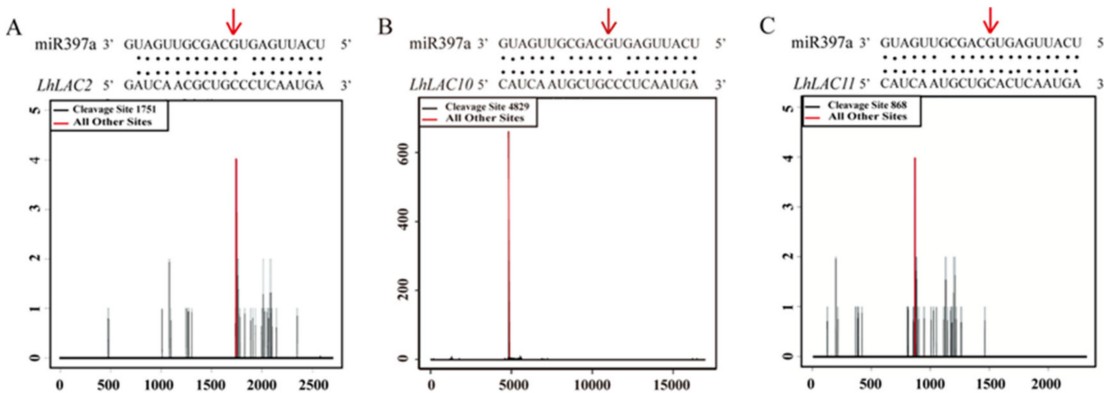

**Figure 2.** Target plot (t-plot) of representative miRNA targets validated by degradome sequencing. The red line and arrow indicate cleavage sites (*x*-axis), and the *y*-axis represent the detected reads. The alignment sequence of miRNA and target are shown on the upper of the t-plots. (**A**) LhmiR397a and its target *LhLAC2*. (**B**) LhmiR397a and its target *LhLAC10*. (**C**) LhmiR397a and its target *LhLAC11*.

To further determine whether LhmiR397a directed the regulation of *LAC11* at the post-transcription level, transient expression of *LhMIR397a* was performed in protoplasts with *LhLAC11* or the mutant of *LhLAC11* (*mLhLAC11*). Sixteen hours after transfection, brilliant green fluorescence was observed when *LhMIR397a* was co-transfected with *mLhLAC11*. When *LhMIR397a* was co-transfected with *LhLAC11*, the fluorescence signal became almost invisible (Figure 3A–F). The green fluorescence signal was also observed when transfected only with *LhLAC11*. The results showed that LhmiR397a could regulate the expression of *LhLAC11* through identification of the complementary sequences. qRT-PCR was used to test the expression of *LhLAC11* in the protoplast transfected with different plasmids. The trend of the expression of qRT-PCR was consistent with the fluorescence signal (Figure 3G). When the plasmid containing *LhLAC11* was delivered into protoplasts, the high expression level of *LhLAC11* was tested. The level of *LhLAC11* transcripts showed a marked decrease when co-transfected with *LhMIR397a*, while a dramatic decline did not appear when *LhMIR397a* was co-transfected with *mLhLAC11*. The results further suggested that *LhLAC11* acts as a direct target of LhmiR397a through complementary sequences.

### 3.3. Downregulation of LAC Transcript Abundance by miR397a Overexpression

To further elucidate the roles of LhmiR397a and its target *LhLAC11*, LhmiR397a precursors *LhLAC11* and *mLhLAC11* were constructed to pBI121-based overexpression plasmids, and then introduced into calli by *Agrobacterium*-mediated transfer. After antibiotic screening culture for 2 months, transgenic calli were observed. Significant overexpression of LhmiR397a was found in randomly selected calli transfected with *LhMIR397a*. To examine whether the expression of *LhLAC11* was altered in *LhMIR397a*-overexpressed calli, the same calli were analyzed by qRT-PCR simultaneously. Consistent with the regulatory pattern of LhmiR397a and its target, the expression of *LhLAC11* was reduced with the increase in LhmiR397a.

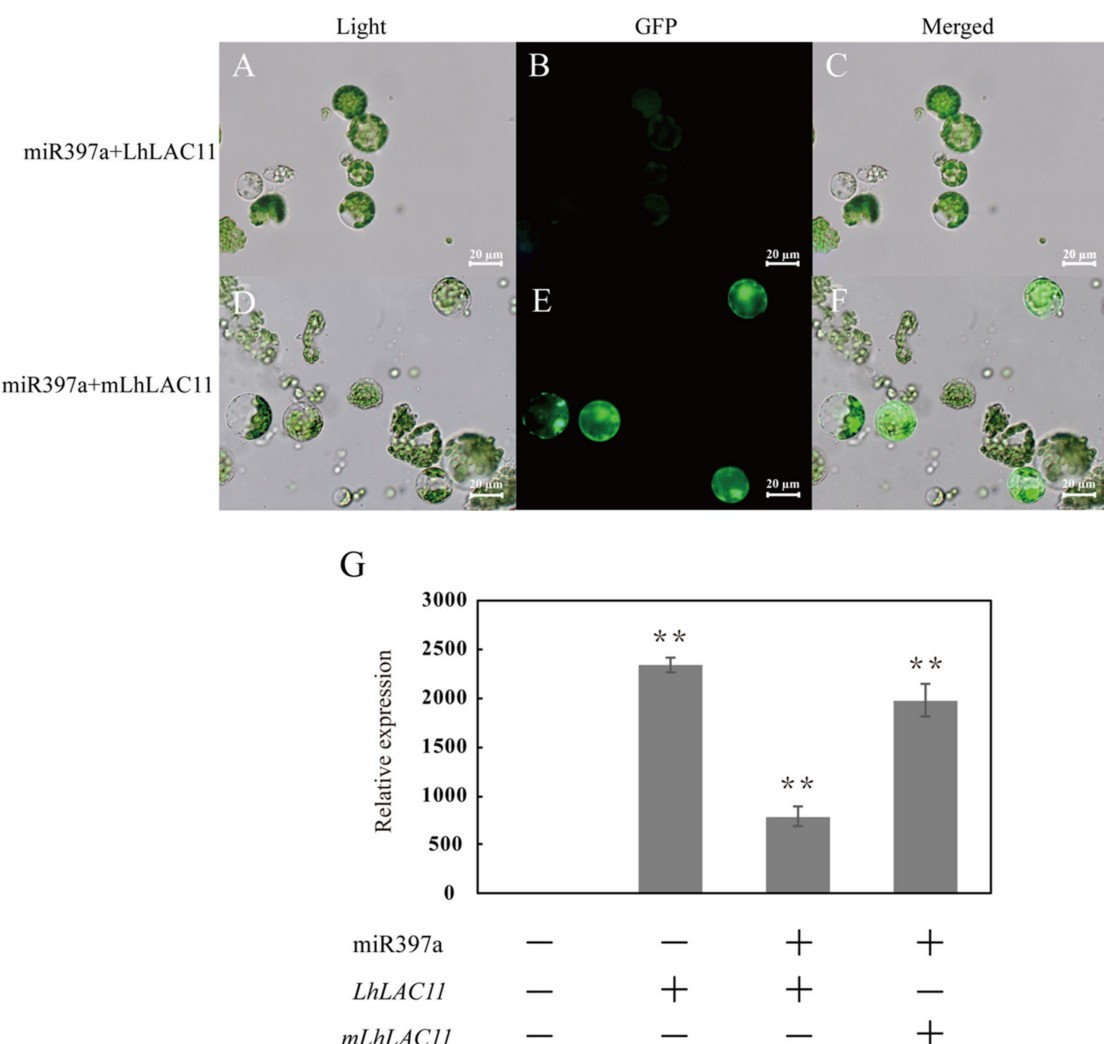

**Figure 3.** Verification that LhmiR397a targets *LhLAC11* by transient expression in protoplasts and qRT-PCR. (**A**–**F**) Transient co-expression of *LhMIR397a* and its target reduces the fluorescence signal. (**A**–**C**) In the 'miR397+*LhLAC11*' group, transient transfection with *LhMIR397a* and its target gene *LhLAC11*. (**D**–**F**) In the 'miR397a+*mLhLAC11*' group, transient transfection with *LhMIR397a* and its target gene the *LhLAC11* mutant (*mLhLAC11*). (**G**) The expression level of *LhLAC11* tested by qRT-PCR. Asterisks indicate statistically significant differences compared with ES1 by Student's t test (** $p \leq 0.01$).

### 3.4. miR397a Downregulated LAC Transcript and Affected Callus Growth and Development

To further elucidate the regulatory role of miR397a-LAC in the callus growth and development process, proliferation was investigated with wild-type and transgenic calli. After culturing for 25 days on the callus-inducing medium, callus proliferation was observed and the proliferation rate of calli, represented by the fresh weight, was analyzed. Callus-overexpressing *LhMIR397a* had a decreased growth rate, and the new growth calli were yellow with small particles. The *LhLAC11*-overexpressing calli had an increased growth rate, and the new callus granules were large and white (Figure 4A–B). Microscopic observation revealed that the calli overexpressing *LhMIR397a* had abundant cell inclusions, while *LhLAC11*- or *mLhLAC11*-overexpressing calli had thin cell contents (Figure 4C).

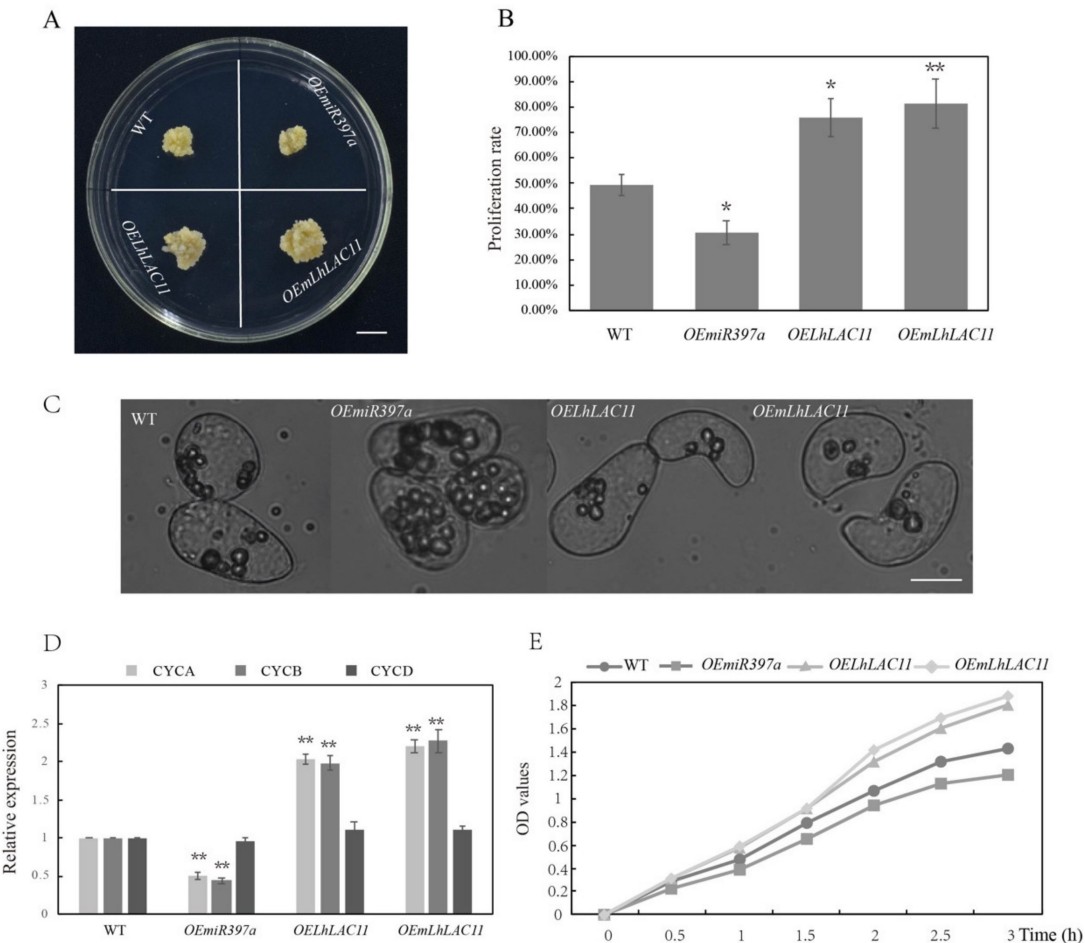

**Figure 4.** Different transgenic calli in *Liriodendron* affect callus proliferation. (**A**) Callus growth of the wild-type (WT), overexpressing miR397a (*OEmiR397a*), overexpressing *LhLAC11* (*OELhLAC11*), and overexpressing *LhLAC11* mutant (*OEmLhLAC11*) on callus-inducing medium after 25 days. Scale bar = 100 μm. (**B**) Callus proliferation displayed as the fresh weight change. Asterisks indicate statistically significant differences compared with wild-type by Student's t test (* *p* < 0.05; ** *p* ≤ 0.01). (**C**) Cells observed with a microscope. (**D**) Expression levels of cell-proliferation-related genes were validated by qRT-PCR. (**E**) MTS assays.

To further investigate the proliferation of calli with different genotypes, qRT-PCR was used to test the expression levels of cell-cycle-related genes, such as cyclin-dependent protein kinase genes. The results showed that the expression levels of *CYCA* and *CYCB* markedly decreased when *LhMIR397a* was overexpressed, while the expression levels increased when *LhLAC11* or *mLhLAC11* were overexpressed. There was no obvious difference in the expression level of *CYCD* (Figure 4D).

Afterwards, callus proliferation was assayed using a tetrazolium compound (3-(4,5-dimethylthiazol-2-yl)-5-(3-carboxymethoxyphenyl)-2-(4-sulfophenyl)-2H-tetrazolium, inner salt [MTS]) and an electronic coupling agent (phenazine ethosulfate). The OD values showed a significant increase in the calli overexpressing *LhLAC11* or *mLhLAC11*, and the OD values decreased in the calli overexpressing *LhMIR397a* (Figure 4E).

In addition to the effect on callus growth, to clarify whether the expression of *LhmiR397a* had an effect on callus development, calli with different genotypes were cultured on somatic embryo-inducing medium, focusing on SE progress. The results showed that the calli overexpressing *LhMIR397a* generated early cotyledons after 21 days of induction, and the somatic embryo induction time was short compared with other genotypes (Figure 5). As abundant cell inclusion is considered to be a favorable condition for SE, we speculated that the overexpression level of LhmiR397a may promote somatic embryo

formation, while the high expression level of *LhLAC11* or *mLhLAC11* may suppress somatic embryo formation.

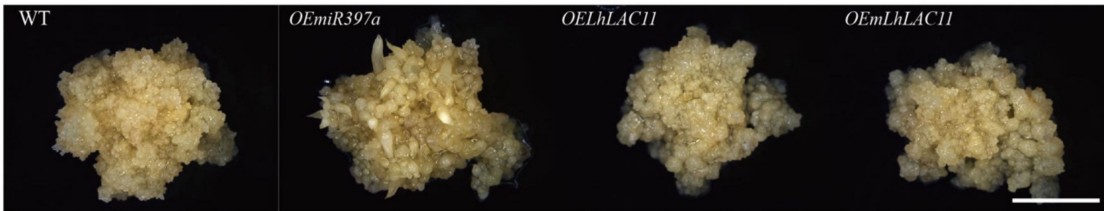

**Figure 5.** Induction of somatic embryo with callus. Transgenic callus with LhMIR397a, LhLAC11, and mLhLAC11 cultured on embryo induction medium after 21 days. Scale bar = 5 mm.

### 3.5. Accumulation of Soluble Sugar, Starch, and Soluble Protein in Callus-Overexpressing miR397a

Concentrations of soluble sugar, starch, and soluble protein were determined in the wild-type and transgenic calli (Figure 6). The results showed that the contents of soluble sugar, starch, and soluble protein significantly increased in the calli overexpressing *Lh-MIR397a*. The amount of soluble sugar in the calli overexpressing *LhMIR397a* was the highest (2.819 mg/g FW); the content was 1.54 times higher than that of the wild-type (1.822 mg/g FW), 1.99 times that of the calli overexpressing *LhLAC11* (1.416 mg/g FW), and 2.19 times that of the calli overexpressing *mLhLAC11* (1.289 mg/g FW). The calli overexpressing *LhMIR397a* had the highest amount of starch (15.231 mg/g FW), and the concentration was 1.5 times higher than that of the wild-type (10.178 mg/g FW), 1.5 times that of the calli overexpressing *LhLAC11* (9.828 mg/g FW), and 1.8 times that of the calli overexpressing *mLh-LAC11* (8.374 mg/g FW). Soluble protein accumulated in the calli overexpressing *LhMIR397a* (35.6 mg/g FW), which was 1.2 times higher than that of the wild-type (30.438 mg/g FW), 1.4 times that of the calli overexpressing *LhLAC11* (25.175 mg/g FW), and 1.6 times that of the calli overexpressing *mLhLAC11* (22.763 mg/g FW).

A

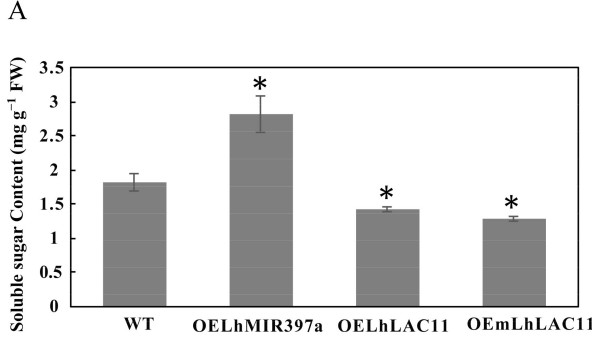

B

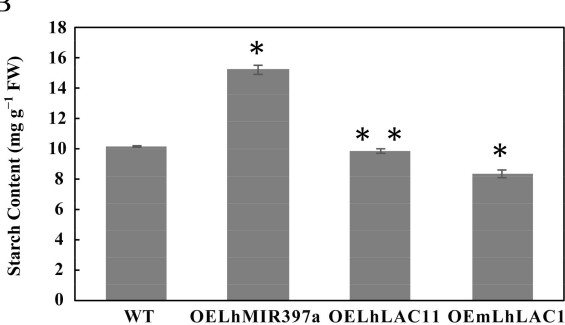

C

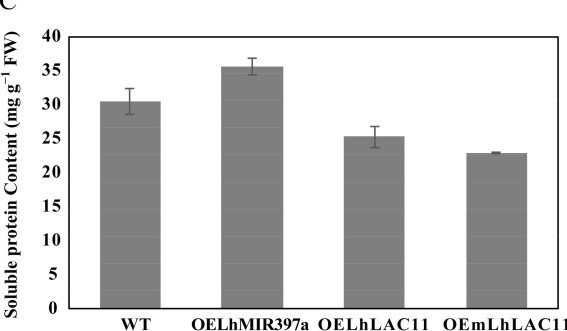

**Figure 6.** Concentration of soluble sugar, starch, and soluble protein in different transgenic calli. (**A**) Soluble sugar concentration. (**B**) Starch concentration. (**C**) Soluble protein concentration. Asterisks indicate statistically significant differences compared with wild-type by Student's t test (* $p < 0.05$; ** $p \leq 0.01$).

## 4. Discussion

### 4.1. Characteristic Analysis of the miR397 Gene and Identification of the Target Transcript

miRNAs are a type of non-coding small RNA with a length of 17–25 nt. High conservation is widespread between the mature miRNA sequences in various plants. miRBase22 was searched, and 84 miR397 members were obtained from 35 plants. The number of miR397 members distributed in the plants was uneven. *Picea abies* had 21 miR397 members, which were the most numerous. *Vitis vinifera*, *Ricinus communis*, *Theobroma cacao*, *Salvia sclarea*, *Digitalis purpurea*, *Nicotiana tabacum*, *Cucumis melo*, *Lotus japonicas*, *Prunus persica*, *Triticum aestivum*, *Fragaria vesca*, and *Camelina sativa* had just one miR397 member. The majority (89%) of miR397 members were 21 nt in length, but 22 and 20 nt miR397 forms were also present. Four miR397 were also identified in *Liriodendron* by BLAST, all of which were 21 nt. After alignment with the miR397 from other plants, the highly conserved sequence was found in the 1st, 2nd, 4th, 14th, and 21st bases, and the bases at positions 5, 7, 9, 10, 11, and 15 were different. Interestingly, miRNAs regulated the cleavage of target transcripts often located between 10 and 11 bases; therefore, the different sites may result in the differentially regulated targets of miRNA family members.

The miRNA genes were transcribed and further processed to generate pre-miRNA, which was in the shape of the stem-loop structure, while the mature miRNAs were located on the arm of the stem-loop [12,44]. The precursor of LhmiR397a could be folded into a stem-loop form with an appropriate ΔG, and the mature LhmiR397a was located on the 5p arm of the precursor. In addition to bioinformatics prediction, cloning was also performed to further identify the sequence of pre-miR397a.

Mature miRNA is incorporated onto an Argonaute (AGO) protein to generate an effector complex called an RNA-induced silencing complex (RISC) [45]. Then, the plant miRNAs are implicated in targeting complementary RNAs for cleavage or perhaps for translational repression within the silencing complex to participate in development regulation [46]. MiR397 was first reported to regulate stress responses in *Arabidopsis* [23] and was then found to be regulated during post-embryogenic development in rice [30]. Afterwards, miR397 was reported to be involved in plant growth, development, and stress response by regulating its target *LAC* gene. In *Liriodendron*, *LhLAC* was predicted as a target of LhmiR397a, and the regulatory role was confirmed by transient transfection with protoplasts. The T-plot analysis further confirmed that *LhLAC* was regulated by LhmiR397a and also showed the cleavage site. We speculated that LhmiR397a performs a regulatory role in *Liriodendron* through its targets.

### 4.2. miR397a-LAC Regulation is Involved in Callus Growth and Development

miRNAs and their targets involved in callus generation and differentiation have been identified in many plants. In rice [30], miR397 exhibited high expression in undifferentiated embryogenic calli (EC), while it had low expression in differentiated tissues. In sweet oranges [17], miR397 was undetectable in EC but showed a high expression in globular-shaped embryos, as well as non-embryogenic calli. The target of miR397a was the *LAC* gene, which is a member of the multicopper-containing oxidase family. The homolog of laccase, ascorbate oxidase [47], was confirmed to be involved in cell growth and may control the cell-division cycle [48]. *SKU*, another member of the multicopper oxidase family, was identified to regulate root development by affecting cell expansion [49,50]. In *Liriodendron*, high expression of LhmiR397a may downregulate the expression of *LhLAC11* and inhibit callus proliferation. When cultured on the callus-inducing medium, the calli genetically modified by *LhMIR397a* overexpression showed a significantly reduced proliferation rate. The calli overexpressing *LhLAC11* had an increased proliferation rate, while *mLhLAC11* overexpression led to a further increase in the proliferation rate. The increase in callus fresh weight in different genotypes may be caused by the cellular morphology or the rate of cell division. Many cyclins are involved in cell-cycle progression. A-type cyclins (*CYCA*) were highly expressed from the S phase to the M phase, and B-type cyclins (*CYCB*) were involved in the G2–M transition and during the M phase; D-type cyclins (*CYCD*) were

mainly present during the G1–S transition [51,52]. In this study, *CYCA* and *CYCB* were downregulated when overexpressing *LhMIR397a* and upregulated when overexpressing the target of *LhLAC11* and *mLhLAC11*, while *CYCD* showed no significant difference. Therefore, miR397a-LAC may affect cell proliferation because it affects the expression of cell-cycle-associated genes.

Studies have shown that the content of soluble sugar, starch, and soluble protein in plants is an important indicator of the physiological and biochemical status of cells and is involved in cell division and differentiation. Soluble sugar is the main energy source of plant growth and development, while starch is considered the main energy storage substance in plant cells, which participates in various physiological and biochemical reactions of the plant body and is an indispensable nutrient in plant culture in vitro. As the product of gene expression, the increase or decrease in protein expression is thought to be related to the process of cell growth and differentiation. In this study, there was a higher content in callus cells overexpressing *LhMIR397a*, and soluble sugar, starch, and soluble protein contents were much higher than in the wild-type. In *Ormosia henryi* [53], the soluble sugar, starch, and soluble protein content were higher in embryogenic calli than in non-embryogenic calli. These phenomena were consistent with the results of *Albizia lebbeck* [54]. The higher the soluble sugar, starch, and soluble protein content in cells, the more active the metabolism in cells. A high soluble sugar, starch, and soluble protein content in *LhMIR397A*-overexpressing callus cells, compared with the wild-type and callus cells overexpressing *LhLac11* or *mLhLac11*, indicated that the *LhMIR397A*-overexpressing callus cells had a better material and energy base for cell differentiation and somatic embryo generation. Consistently, we found that the calli overexpressing *LhMIR397a* exhibited a short time for somatic embryo induction compared with the wild type.

## 5. Conclusions

In summary, four miR397s were identified in *L. sino-americanum,* and *LhLAC* was confirmed as a target of *LhmiR397a*. The high expression of LhmiR397a resulted in an decreased expression level of *LhLAC* and then inhibited callus proliferation and showed a significantly reduced proliferation rate. The calli overexpressing *LhMIR397a* showed a higher content of soluble sugar, starch, and soluble protein and required a shorter time for inducing somatic embryos than the wild-type. This provides us with a deep understanding of the regulatory function of miR397a-LAC in callus growth and development in *Liriodendron*. It also provides a useful resource for further studies on the genetic mechanisms of callus growth and development in other plants, since *Liriodendron* is regarded as a basal angiosperm.

**Author Contributions:** D.W., J.S. and J.C. conceived and designed the research. D.W. performed the experiments. D.W., J.C. performed data analysis. F.L., Y.L., T.C., Z.H. contributed reagents/materials. D.W., J.S. and J.C. wrote the manuscript. All authors have read and agreed to the published version of the manuscript.

**Funding:** This work was supported by the Natural Science Foundation of China [32071784], the State Forestry and grassland administration [2020133106], and the Priority Academic Program Development of Jiangsu Higher Education Institutions (PAPD).

**Conflicts of Interest:** The authors declare that they have no conflict of interest to report regarding the present study.

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
