# Peer review of "Identification of miR397a and Its Functional Characterization in Callus Growth and Development by Regulating Its Target in Liriodendron"

_forests, doi:10.3390/f12070912_

Round 1
Reviewer 1 Report
In this manuscript, it was shown the regulatory function of miR397 during callus growth and development in Liriodendron. LhmiR397a and its targets were identified. The regulatory role of LhmiR397a and LhLAC11 was demonstrated using qRT-PCR and transient expression in protoplasts.
The manuscript provided new knowledge of how miR397 controls callus growth.
I have the following comments:
- Materials and Methods
2.3. Cloning and transformation:
-Missing the reference for primers.
- The manufacturer's data for the pJIT166 plasmid is missing.
2.4. Quantitative real-time PCR
Provide the reference or the data about qRT-PCR reaction (cycling conditions, controls...)
- Results
Subchapter 3.3. Cloning and overexpression of the precursor miR397a was already described in Materials and methods and does not belong to the result. In my opinion, the sentence could be reformulated and integrated into subchapter 3.4.
- Conclusion
The conclusion could be improved. It should be written in such a way that it demonstrates the data derived from the study. In my opinion, the sentences "According to previous studies, miR397 plays a role in callus development, while the regulatory function of miR397a-LAC in Liriodendron was not exact. To clarify this, pre-miR397a was cloned, and transgenic plants were obtained with an Ag-robacterium-mediated approach" should be reformulated.
Author Response
Thank you for your comments and the conclusions has been changed.
Point 1: In the part of “Materials and Methods: Cloning and transformation” Missing the reference for primers
Response 1: Thank you for reminding us the missing the reference for primers. We have added the primers of the mutagenic primer for mutant LhLAC11and the qRT-PCR primers for miR397, LhLAC11, CYCA, CYCBand CYCD.
Point 2: The manufacturer's data for the pJIT166 plasmid is missing.
Response 2: Thank you for the comment. The vector was provide by Dr. Ailing Huo (Nanjing Forestry University). And the manufacturer's data for the pJIT166 plasmid was described by Huo et al (2017). pJIT166-GFP is a high-copy vector driven by a double 35S cauliflower virus promoter. It harbors the GFP gene and a terminal nopaline synthase sequence. Ampicillin served as the bacterial selection marker.
Point 3: In the part of Quantitative real-time PCR, Provide the reference or the data about qRT-PCR reaction (cycling conditions, controls...)
Response 3: Thank you for the comment. qRT-PCR was performed as described by Li et al (2012). The expression level of 18S and Actin were used as the internal control to standardize the RNA samples for each reaction. The relative gene expression levels were calculated using the 2−△△CTmethod, and error bars indicate the standard deviation of three sample replicates.
Point 4: In the part of “Result”, Subchapter 3.3. Cloning and overexpression of the precursor miR397a was already described in Materials and methods and does not belong to the result. In my opinion, the sentence could be reformulated and integrated into subchapter 3.4.
Response 4:Thank you for reminding us the improper description on the study. We have deleted the subchapter 3.3, and the sentence has been reformulated and integrated into subchapter 3.4.
Point 5: The conclusion could be improved. It should be written in such a way that it demonstrates the data derived from the study. In my opinion, the sentences "According to previous studies, miR397 plays a role in callus development, while the regulatory function of miR397a-LAC in Liriodendron was not exact. To clarify this, pre-miR397a was cloned, and transgenic plants were obtained with an Agrobacterium-mediated approach" should be reformulated.
Response 5: Thank you for reminding us the improper description on the study. We have deleted the sentences "According to previous studies, miR397 plays a role in callus development, while the regulatory function of miR397a-LAC in Liriodendron was not exact. To clarify this, pre-miR397a was cloned, and transgenic plants were obtained with an Agrobacterium-mediated approach", and reformulated the conclusion.

Reviewer 2 Report
Editing suggestions
General: would be good to have something in the introduction about why the study is done. Why is the miRNA function in Liriodendron worth investigating? Why should we be interested in the results? What is the larger context of this study: are you trying to determine the function of this miRNA in many different plants, or investigating a callus development pathway in this tree species specifically?
- There is a consistent error with the scientific name Liriodendron chinense appearing as Liriodendron Chinese. Please fix all occurences.
- 2.2 Bioinformatics analysis: software should be cited if possible rather than just providing web link
- Figures 4 and 6: You should provide some indication of what the asterisks mean in the caption of the figures. I assume they indicate significant differences; if so, you need to briefly mention what test was used.
Author Response
Thank you for your comments, and we have revised the paper according to the review comments.
Point 1:
1、Would be good to have something in the introduction about why the study is done.
Response:Previous studies have demonstrated that miRNAs are essential for cell differentiation and the development process during early embryogenesis. miR397 were broadly involved in plant growth, development, and stress response. In our previous study, we found that miR397 can be tested by chip hybridization, and the targeted transcripts of miR397 were identified by the degradome sequence. However, the regulatory effect of miR397 during callus development and differentiation inLiriodendron sino-americanumremains unknown.
2、Why is the miRNA function in Liriodendron worth investigating? Why should we be interested in the results?
Response:Liriodendron, a member of the Magnoliaceae, is regarded as a basal angiosperm. It is an excellent woody tree species and garden afforestation tree species. Therefore, it is crucial to improve the quality and yield of Liriodendron sino-americanum. In view of previous studies, the somatic embryogenesis system represents a very powerful tool for massive propagation and trait improvement of higher plants. As a result, determination of the mechanism of somatic embryogenesis in Liriodendron sino-americanumis urgently needed. And the regulatory roles of miRNAs during early somatic embryogenesis in Liriodendron sino-americanumprovided a useful resource for further studies on the genetic mechanisms during SE in other plants.
3、What is the larger context of this study: are you trying to determine the function of this miRNA in many different plants, or investigating a callus development pathway in this tree species specifically?
Response:The larger context of this study are trying to investigating a callus development pathway in this tree species specifically. Callus growth and development, a crucial process in plant propagation, is involved in abundant gene regulation. MiRNAs are key regulators in the process of cell differentiation and development. MiR397 was identified to participate in plant growth and development and respond to stress. While the regulatory function of miR397 during callus growth and development was not clear in Liriodendron. This study identified LhmiR397a, and provided a functional characterization of LhmiR397a in callus growth and development by regulating its target in Liriodendron.
Point 2: There is a consistent error with the scientific name Liriodendron chinense appearing as Liriodendron Chinese. Please fix all occurences.
Response 2: Thank you for reminding us the improper description on the study. We have revised to Liriodendron chinense.
Point 3: Bioinformatics analysis: software should be cited if possible rather than just providing web link.
Response 3: Thank you for reminding us the improper description on the study. We have revised and added the reference about the software.
Point 4: Figures 4 and 6: You should provide some indication of what the asterisks mean in the caption of the figures. I assume they indicate significant differences; if so, you need to briefly mention what test was used.
Response: Thank you for reminding us the improper description on the study. We have revised.Asterisks indicate statistically significant differences compared with ES1 by Student’s t test (**P ≤0.01). Asterisks indicate statistically significant differences compared with wild type by Student’s t test (*P < 0.05).
